# Early and Late Transcriptomic and Metabolomic Responses of *Rhododendron* ‘Xiaotaohong’ Petals to Infection with *Alternaria* sp.

**DOI:** 10.3390/ijms241612695

**Published:** 2023-08-11

**Authors:** Xi-Min Zhang, Jie-Ting Li, Ying Xia, Xiao-Qian Shi, Xian-Lun Liu, Ming Tang, Jing Tang, Wei Sun, Yin Yi

**Affiliations:** 1Key Laboratory of Plant Physiology and Development Regulation, Guizhou Normal University, Guiyang 550025, China; lqa1678@163.com (J.-T.L.); 13595747814@163.com (Y.X.); 222100100442@gznu.edu.cn (X.-Q.S.); tangjing2016@gznu.edu.cn (J.T.); sunwei889@163.com (W.S.); gzklppdr@gznu.edu.cn (Y.Y.); 2Key Laboratory of Environment Friendly Management on Alpine Rhododendron Diseases and Pests of Institutions of Higher Learning in Guizhou Province, Guizhou Normal University, Guiyang 550025, China; llunxian@163.com; 3School of Life Sciences, Guizhou Normal University, Guiyang 550025, China; mingtang@gznu.edu.cn; 4Key Laboratory of State Forestry Administration on Biodiversity Conservation in Karst Area of Southwest, Guizhou Normal University, Guiyang 550025, China

**Keywords:** rhododendron variety, fungus, petal blight disease, hydrogen peroxide, defense-related genes, phytoalexin

## Abstract

In recent years, petal blight disease caused by pathogens has become increasingly epidemic in *Rhododendron*. Breeding disease-resistant rhododendron is considered to be a more environmentally friendly strategy than is the use of chemical reagents. In this study, we aimed to investigate the response mechanisms of rhododendron varieties to petal blight, using transcriptomics and metabolomics analyses. Specifically, we monitored changes in gene expression and metabolite accumulation in *Rhododendron* ‘Xiaotaohong’ petals infected with the *Alternaria* sp. strain (MR-9). The infection of MR-9 led to the development of petal blight and induced significant changes in gene transcription. Differentially expressed genes (DEGs) were predominantly enriched in the plant–pathogen interaction pathway. These DEGs were involved in carrying out stress responses, with genes associated with H_2_O_2_ production being up-regulated during the early and late stages of infection. Correspondingly, H_2_O_2_ accumulation was detected in the vicinity of the blight lesions. In addition, defense-related genes, including PR and FRK, exhibited significant up-regulated expression during the infection by MR-9. In the late stage of the infection, we also observed significant changes in differentially abundant metabolites (DAMs), including flavonoids, alkaloids, phenols, and terpenes. Notably, the levels of euscaphic acid, ganoderol A, (−)-cinchonidine, and theophylline in infected petals were 21.8, 8.5, 4.5, and 4.3 times higher, respectively, compared to the control. Our results suggest that H_2_O_2_, defense-related genes, and DAM accumulation are involved in the complex response mechanisms of *Rhododendron* ‘Xiaotaohong’ petals to MR-9 infection. These insights provide a deeper understanding of the pathogenesis of petal blight disease and may have practical implications for developing disease-resistant rhododendron varieties.

## 1. Introduction

Ericaceae plants, comprising a diverse family of flowering plants, are known for their clusters of distinctly colored flowers, making them popular for urban and garden greening purposes [1]. Among these, the Baili Azalea Nature Reserve (BANR) in the northwest Guizhou province of China is renowned for its extensive natural azalea forest, attracting visitors from around the world and serving as a crucial tourism resource for local communities [2]. However, the spread of petal blight disease caused by pathogens has become increasingly prevalent in recent years, particularly during the high temperature and rainy periods from March to June. The disease progresses through the appearance of small brown spots on infected petals, eventually spreading throughout the entire petal, leading to wilting and even falling. Petal blight not only shortens the flowering period and reduces ornamental value but also hampers seed development and poses a significant economic threat to the tourism industry, as well as diminishing species diversity in the ecosystem.

Currently, the application of chemical fungicides such as Parzate, Dithane, or Z-78 has been proven to be an effective method for preventing petal blight disease in rhododendron species [3]. However, the excessive use of fungicides can lead to drug resistance in the pathogens and severe environmental pollution from residual fungicides [4,5]. Therefore, developing disease-resistant rhododendron varieties through screening and breeding is crucial. Moreover, comprehending the molecular mechanism underlying petal blight disease in rhododendron species is essential for cultivating disease-resistant varieties and promoting environmentally friendly control methods.

When plants are infected by pathogens, the immune system is triggered through pattern recognition receptors (PRRs) located on the cell membranes. These PRRs recognize pathogen-associated molecular patterns (PAMPs), initiating PAMP-triggered immunity (PTI) [6]. Moreover, nucleotide-binding leucine-rich repeat receptors (NLRs) recognize, either directly or indirectly, effector molecules secreted by the pathogen, instigating the second immune system, which is known as effector-triggered immunity (ETI) [6]. These innate immune systems work cooperatively, producing downstream responses such as the biosynthesis of plant hormones, the production of reactive oxygen species (ROS), the accumulation of secondary metabolites, the induction of stomatal closure, the triggering of hypersensitive response (HR), and the activation of defense-related gene expression [7,8,9,10].

Plant secondary metabolites play a crucial physiological function in resisting pathogen infection [11,12]. Upon pathogen infection, an accumulation of secondary metabolites occurs in the tissues surrounding the infection site, such as flavonoids, terpenes, and nitrogen/sulfur compounds, or in the infection site (such as coumarins) [13,14], all accumulations possessing the ability to eliminate, hinder the growth of, and prevent the spread of pathogens [15], thus effectively reducing the impact of diseases caused by pathogens. The induction of secondary metabolites by pathogens has been reported in various plant species. For instance, the infection of rice by the rice blast fungus stimulates the induction of hesperidin [16], while *Alternaria alternata* induces the accumulation of scopoletin, scopolin, and capsaicin in tobacco [17,18,19], and Arabidopsis infected with Botrytis cinerea induces the accumulation of camalexin [20]. Similarly, cabbage infected with *Alternaria brassicae* accumulates 4-methylindole-3-methylthioglucoside [21]. The specific secondary metabolites induced by the infection of *A. alternata* in *Rhododendron* ‘Xiaotaohong’ petals, however, are still unclear.

Transcriptomics is a valuable tool for identifying genes that are activated in plants after the plants are infected by pathogens, providing insights into disease resistance-related genes [22]. In contrast, metabolomics is used for analyzing changes in metabolite levels in infected plants [23]. The combination of transcriptomics and metabolomics has been used to analyze plant disease resistance genes and metabolites in various crops, such as maize [24], potato [25], rice [26], and bean [27]. However, there is a scarcity of research reports on *Rhododendron* ‘Xiaotaohong’ petals in the context of transcriptomics and metabolomics analysis, warranting further research in this area.

In a previous study, we successfully isolated and identified the primary fungus (*Alternaria* sp.) from wild diseased *Rhododendron delavayi* petals, verifying that this fungus can cause petal blight disease by infecting petals. Based on these findings, the present study aims to investigate key factors involved in the response of petals to *Alternaria* sp. infection, using transcriptomics and metabolomics. This research will provide valuable insights into the underlying mechanisms of petal response to fungal infection and serve as a foundation for further studies in this field.

## 2. Results

### 2.1. Alternaria sp. Infection Caused the Petal Blight

The *Rhododendron* ‘Xiaotaohong’ unpunctured petals remained healthy for a period of 144 h without any treatment, as demonstrated in Figure 1A. In the mock treatment, no blight symptoms were observed on the punctured petals for 144 hpi. However, the punctured petals infected with MR-9 displayed brown blight spots surrounding the wound after 12 hpi (Figure 1A). The lesion area was considerably greater after 12 hpi than that of the mock, and the largest lesion area observed was up to 44.87 mm^2^ in 144 hpi (Figure 1B). Consequently, we elected to analyze the transcriptome and metabolome at 12 h as the early stage and 144 h as the late stage of infection.

### 2.2. RNA-Seq, Analysis of DEGs, and qRT-PCR Validation

To investigate the early and late response mechanisms of *Rhododendron* ‘Xiaotaohong’ petals to MR-9 infection, sequencing was carried out on both infected and mock petals at 12 and 144 hpi utilizing the Illumina platform. Each sample produced an excess of 40 million clear reads with a guanine and cytosine/total base (GC content) of approximately 50%, and a percentage of base calling accuracy quality at 99.9% (Q30) of not less than 94.42%. The clean reads were aligned to the reference genome, with a matching rate ranging from 81.97% to 85.27% (Appendix A).

To visualize the variation and similarity of all the samples, we performed principal component analysis (PCA) on normalized fragments per kilobase of transcript per million mapped reads (FPKM) values of all detectable genes at different times for the MR-9 infected and mock petals. PC1 accounted for 28.84%, and PC2 accounted for 16.50% of the total variation (Figure 2A). The results demonstrated that the data from the three repeated treatments were closely clustered with strong repeatability, and the variation of samples between treatment groups was minimal. Moreover, following MR-9 infection, all detected genes were isolated from each other, implying that various transcriptional changes were induced in the petals (Figure 2A).

We conducted an analysis to discern differentially expressed genes (DEGs) between *Rhododendron* ‘Xiaotaohong’ petals infected with MR-9 and mock-infected controls at each time point. Our results indicate a considerable number of DEGs during 12 h post-infection (hpi) (1515 up-regulated, 903 down-regulated) and 144 hpi (749 up-regulated, 424 down-regulated) (Figure 2B), when compared to mock controls. Thus, our findings suggest that the fungal infection by MR-9 substantially triggers transcriptional responses in *Rhododendron* ‘Xiaotaohong’ petals. This underscores the notable impact of MR-9 infection on the gene expression profile of *Rhododendron* ‘Xiaotaohong’ petals.

In order to increase the dependability of the differentially expressed genes (DEGs) identified via RNA sequencing, we chose eight genes with diverse expression levels at 12 and 144 h post-infection (hpi) for further analysis employing quantitative real-time PCR (qRT-PCR). Consistent with our expectations, the qRT-PCR results were in concurrence with the RNA-seq results (y = 0.996x − 0.443, R^2^ = 0.555) (Appendix A). This strongly supports the trustworthiness of RNA-seq for gene expression analysis.

### 2.3. KEGG and GO Enrichment by DEGs in Early and Late Infection

We conducted further analysis of biological functions associated with the differentially expressed genes (DEGs) through enrichment analysis using *The Kyoto Encyclopedia of Genes and Genomes* (KEGG) and *Gene Ontology* (GO). In the KEGG enrichment pathway analysis for up-regulated DEGs at 12 hpi, 19 pathways showed significant enrichment (*p* < 0.05), including “plant–pathogen interaction”, “phenylpropanoid biosynthesis”, “MAPK signaling pathway-plant”, and “plant hormone signal transduction” (Figure 3A) (Appendix A). Similarly, up-regulated DEGs at 144 hpi showed significant enrichment in 13 pathways (*p* < 0.05), such as “plant–pathogen interaction”, “phenylpropanoid biosynthesis”, “MAPK signaling pathway-plant”, and “plant hormone signal transduction” (Figure 3C) (Appendix A). In the GO enrichment analysis, the up-regulated DEGs identified at 12 hpi and 144 hpi were assigned to three major GO classes, namely biological process, cellular component, and molecular function, with 2034, 235, and 297 categories (Appendix A) and 1850, 283, and 540 categories (Appendix A), respectively. The top eight categories with significantly enriched DEGs were reported (Figure 3B,D). It is worth noting that the GO terms “response to stimulus”, “response to stress”, and “response to fungus” were significantly enriched both at 12 hpi and 144 hpi (Figure 3B,D).

The majority of differentially expressed genes (DEGs) were found to be enriched in plant–pathogen interaction pathways, with 126 DEGs at 12 hpi and 46 DEGs at 144 hpi (Appendix A) (Figure 3A,C). Figure 4A displays the pathway of reactive oxygen species (ROS) production and some identification of DEGs were involved in fungal infection in the *Rhododendron* ‘Xiaotaohong’ petals. Among the DEGs, two DEGs were annotated as disease resistance protein (CF9), three DEGs as calcium-dependent protein kinase (CDPK), two DEGs as respiratory burst oxidase (Rboh), and four DEGs as cyclic nucleotide gated channel (CNGC), all exhibiting a high expression level at either 12 or 144 hpi (Figure 4B). Hydrogen peroxide (H_2_O_2_) staining analysis revealed that H_2_O_2_ was distributed around the blight spots, and the content in infected petals was notably higher than in mock controls (Appendix A). The regulation of defense-related genes in the plant–pathogen interactions pathway is depicted in Figure 4C. Four DEGs were annotated as LRR receptor-like serine/threonine protein kinase FLS (FLS2), two DEGs as somatic embryogenesis receptor kinase 4 (SERK4), six DEGs as brassinosteroid-insensitive 1-associated receptor kinase 1 (BAK1), one DEG as LRR receptor-like serine/threonine-protein kinase EFR (EFR), one DEG as mitogen-activated protein kinase (MAPK), seven DEGs as WRKY transcription factor 33 (WRKY33), four DEGs as WRKY transcription factor 22 (WRKY22), one DEG as WRKY transcription factor 29 (WRKY29), three DEGs as senescence-induced receptor-like serine/threonine protein kinase (FRK1), and four DEGs as pathogenesis-related protein (PR1). The expression changes of these DEGs were shown in the Figure 4D.

### 2.4. Major Transcription Factor (TF) Families Were Activated in Early and Late Infection

Transcription factors (TFs) are crucial for the transcriptional reprogramming of plants under biotic stress. To identify TF families in response to MR-9 infection, we analyzed the differentially expressed genes (DEGs) annotated as TFs in infected petals. Our analysis revealed 192 up-regulated DEGs predicted to encode 20 TF families at 12 hpi, with the RLK-Pelle_DLSV, WRKY, and AP2/ERF-ERF families being the most dominant (Figure 5A). Similarly, 62 up-regulated DEGs were predicted to encode 20 TF families at 144 hpi, with the RLK-Pelle_DLSV, AP2/ERF-ERF, WRKY, and MYB families being the most prevalent (Figure 5B). In addition, Appendix A shows the predicted TFs encoded by down-regulated DEGs.

### 2.5. Metabolite Identification and Differentially Abundant Metabolites (DAMs) Analysis

The UHPLC–MS platform, in combination with annotation databases, was utilized to identify metabolites in both infected and mock-treated petals. The UHPLC–MS chromatograms of the 12 samples obtained from infected and mock treatment petals exhibited good reproducibility, signifying the stability and reliability of the run conditions and the instrument. The relative standard deviation (RSD) of the internal standard (2-chorophenylalanine) in the quality control sample was measured to be 6.42%, indicating the system’s stability. A total of 884 peaks were extracted, and 811 metabolites were tentatively identified (Appendix A).

We employed principal component analysis (PCA) to assess the similarities and differences in metabolite levels between infected and mock petals. The analysis indicated that the first four components explained 48.9% of the total variation in metabolite levels (Appendix A). PC1 and PC2 accounted for 14.9% and 14.2% of the total variation, respectively (Figure 6A). These results indicate that there are significant differences in metabolites between infected and mock control petals.

Variable importance in projection (VIP) and Student’s *t*-test were employed to identify the differentially abundant metabolites (DAMs) present in infected and mock petals at the same time point (VIP > 1, *p* < 0.05). A substantial number of DAMs were identified in 12 hpi (14 up-regulated, 63 down-regulated) and 144 hpi (51 up-regulated, 25 down-regulated) (Figure 6B). The 14 up-regulated DAMs identified in 12 hpi consisted of three flavonoids, two phytohormones, two terpenes, and seven others (Figure 6C). The 51 up-regulated DAMs identified in 144 hpi included nine flavonoids, six terpenes, six alkaloids, five phenols, and others (Figure 6D). A hierarchical clustering method is used to display 26 up-regulated DAMs in mock and infected petals at 144 hpi (Figure 6E). Additionally, a correlation analysis was conducted between differentially expressed genes and metabolites. The results showed that many up-regulated genes were positively correlated with the accumulation of H_2_O_2_ (as a metabolite) at 12 h, and that gentisic acid, karanjin, ganoderol A, and euscaphic acid showed positive correlation at 144 h (Appendix A). Collectively, the outcomes suggest that the fungal MR-9 infection resulted in substantial changes in metabolites in petals.

### 2.6. Differentially Abundant Metabolites Related to Plant–Pathogen Interactions

Flavonoids, alkaloids, terpenes, and phenols are among the most common secondary metabolites known to exhibit antifungal activity. In response to fungal infection, 11 metabolites (Cinchonine, (−)-cinchonidine, theophylline, ganoderic acid, naringenin, gentisic acid, luteolin, karanjin, linarin, ganoderol A, and euscaphic acid) showed a substantial increase, over threefold, compared to the control (Table 1). Notably, six of these metabolites (naringenin, gentisic acid, luteolin, vestitol, karanjin, and euscaphic acid) have been extensively documented as possessing antifungal activity in other species (Table 1).

## 3. Discussion

The innate immunity of plants primarily involves the identification of conserved PAMPs by PRRs and effector recognition by NLRs. The pathogen infection signals are transmitted to the cytoplasm, thereby activating mitogen-activated protein kinase (MAPK) cascades and other signaling transduction pathways. MAPK cascades play a crucial role in signaling diverse defense responses, such as hormone biosynthesis, reactive oxygen species (ROS) generation, stomatal closure, defense gene activation, phytoalexin biosynthesis, and hypersensitivity response (HR) [9].

### 3.1. Reactive Oxygen Species Were Involved in Response to MR-9 Infection

Petal blight caused by the blight fungus MR-9 was observed in *Rhododendron* ‘Xiaotaohong’ petals 12 h post-infection (hpi), with the lesion area being at its largest at 144 hpi of the 144 hpi experiment (Figure 1). This suggests that the fungus has the ability to infect and cause petal blight. Transcriptomic analysis revealed a significant number of differentially expressed genes (DEGs) in the infected petals, with more DEGs identified at 12 hpi than at 144 hpi (Figure 2). This observation indicates that *Rhododendron* ‘Xiaotaohong’ petals rapidly activate gene transcription in response to fungal infection in the early stages. In addition, a difference between mock 12 hpi and mock 144 hpi was observed (Figure 2A), suggesting that the reaction is caused by puncture. Further analysis revealed that the most annotated DEGs were enriched in the plant–pathogen interaction pathway when the petals were infected by MR-9, and these DEGs mainly performed the biological function of responding to stimuli and stress, as determined through *Gene Ontology* (GO) enrichment analyses (Figure 3). These results indicate that genes activated by fungal infection mainly function in defense against MR-9 infection.

Both PTI and ETI defense mechanisms can rapidly induce the production of ROS in plant cells, such as H_2_O_2_ [34], which can trigger hypersensitivity response (HR) and local cell death, thereby preventing pathogen colonization [27,35,36]. An increase in cytoplasmic Ca^2+^ is a crucial early event in the response to pathogen signaling cascades, primarily enabled by Ca^2+^ influx into the cytoplasm through cyclic nucleotide gated channels (CNGC) present on the plasma membrane [37,38]. Calcium-dependent protein kinases (CDPKs), key downstream components of Ca^2+^ signaling, play a significant role as Ca^2+^ sensors and mediate the synthesis of ROS in response to the signal cascade of plant–pathogen interactions. In the current study, four up-regulated genes encoding CNGC and three up-regulated genes encoding CDPK were activated after fungal infection (Figure 4), consistent with previous reports in plants [39,40,41]. After MR-9 infection, the gene encoding NADPH oxidase was up-regulated (Figure 4) and this was accompanied by H_2_O_2_ accumulation (Appendix A), suggesting that the Ca_2+_ signaling pathway activated CDPK expression and induced ROS production in the early defense of petals against MR-9 infection [42]. Furthermore, correlation analysis between differentially expressed genes and metabolite accumulation showed that a considerable number of genes were positively correlated with H_2_O_2_ accumulation during the early stages (Appendix A), thereby indicating the vital role of H_2_O_2_ in early defense.

### 3.2. Defense-Related Genes Played an Important Role in Response of MR-9 Infection in Petals

Plants activate defense-related genes in response to pathogen infection, which is a crucial mechanism for both PTI and ETI [43]. Notably, various defense-related genes participate in the response of plants to fungal infections, including genes for secondary metabolite synthesis and pathogenesis-related proteins (PR). In the plant–pathogen interaction pathway, transcriptome analysis of *Rhododendron* ‘Xiaotaohong’ petals infected with MR-9 revealed numerous up-regulated DEGs in both early and late stages, which were annotated as defense-related genes, such as senescence-induced receptor-like serine/threonine-protein kinase (FRK) and PR. These defense genes potentially play a direct role in the response of *Rhododendron* ‘Xiaotaohong’ petals against MR-9 infection. Comparable observations have been reported in *Medicago truncatula*, rice, and lemon [42,44,45].

### 3.3. Transcription Factors (TFs) Play an Important Role in Response to MR-9 Infection

Increasing evidence suggests that several transcription factor (TF) families, such as ERF, bZIP, bHLH, WRKY, NAM, ATAF, CUC (NAC), and MYB, play a critical role in innate immunity in plants [46,47]. These TFs function by mediating and regulating the expression of defense-related genes, thereby enhancing plant resistance against pathogen attacks during both PTI and ETI. Previous studies have identified several major TF families, such as WRKY, AP2/ERF, bZIP, bHLH, and MYC, which are involved in plant defense response [48,49,50,51]. In the current study, we observed a significant upregulation of the RLK, WRKY, AP2/ERF, and MYB families following pathogen infection (Figure 5), suggesting their potential involvement in the response to MR-9 infection. Specifically, five upregulated genes encoding WRKY were identified to be involved in the plant–pathogen interaction pathway. This finding points to the direct participation of the WRKY families (WRKY22, WRKY29, and WRKY33) in the positive regulation of defense-related gene expression, consistent with prior reports [52,53]. Furthermore, some TFs may also play a role in the negative regulation of defense-related gene expression during the plant–pathogen interaction (Appendix A).

### 3.4. Accumulation of Antifungal Secondary Metabolites Contributed to the Defense Response of MR-9 Infection in Petals

Plant-specific secondary metabolites are essential for the defense response of plants [54]. These metabolites, such as alkaloids, flavonoids, phenols, terpenes, and anthocyanin, have been shown to possess potent anti-pathogen properties [12,13,55,56]. Our study observed a significant increase in the accumulation of nine flavonoids, six alkaloids, five phenols, and six terpenes in petals infected with MR-9 in the late stage (Figure 6D), indicating that these metabolites may play a critical role in the response to MR-9 infection.

Phytoalexins are low-molecular-weight compounds generated by plants in response to pathogen attack, and they exhibit anti-pathogen activities [57]. Various secondary metabolites, including naringenin [28], luteolin [29], gentisic acid [32], vestitol [30], karanjin [31], and eucalyptic acid [33], have been documented to possess antifungal activity in crops. In our study, we observed a dramatic accumulation of euscaphic acid (21.8 times), ganoderol A (8.5 times), (−)-cinchonidine (4.5 times), and theophylline (4.3 times) as secondary metabolites in MR-9-infected petals (Table 1). Notably, euscaphic acid and ganoderol A were positively correlated with several gene expressions (Appendix A), suggesting their potential as phytoalexins in *Rhododendron*. Future studies utilizing genetic methods and antifungal analyses could confirm their roles in plant defense.

In summary, KEGG annotation of DEGs indicated that the expression of genes producing H_2_O_2_ was up-regulated in both the early and late stages of infection, along with the accumulation of H_2_O_2_ around the blight spots. Moreover, up-regulation of defense-related genes such as PR1 and FRK1 was also observed. During the later stages of MR-9 infection, numerous DAMs were identified, including flavonoids, alkaloids, phenols, and terpenes. The accumulations of euscaphic acid, ganoderol A, (−)-cinchonidine, and theophylline in infected petals were higher than those in control petals, respectively (Figure 7). These metabolites can be suggested as potential phytoalexins to be further investigated using genetic methods and antifungal analysis.

## 4. Materials and Methods

### 4.1. Plant Materials and Alternaria sp. Strains

*Rhododendron* ‘Xiaotaohong’ seedlings were propagated using cutting techniques from a stock plant in a flower nursery situated in Huishui County, Guizhou Province, China. Mature 3-year-old seedlings with flowers were transferred to a controlled greenhouse environment (400 µmoL quanta m^2^/s, 25 °C, 12 h light and 12 h dark) at the Key Laboratory of Plant Physiology and Development Regulation, Guizhou Normal University. The *Alternaria alternate* sp. strain (MR-9), which is responsible for petal blight disease, was isolated from *R. delavayi* exhibiting petal blight disease [58] and is presently maintained in the Key Laboratory of Plant Physiology and Development Regulation.

### 4.2. Fungal Infection and Disease Assays

*Rhododendron* ‘Xiaotaohong’ flowers in full bloom were selectively chosen to undergo inoculation with MR-9, utilizing established laboratory methods [59]. Petals were punctured with a sterilized 2 mm carving knife, and a mycelial suspension was prepared and adjusted to a concentration of 0.025 mg/mL using sterilized H_2_O. A total of 7 μL of mycelial suspension were added to the punctured petal wounds for fungal inoculation, while sterilized H_2_O was used as a mock treatment. The fungal-inoculated petals were then covered with plastic bags and maintained in a growth house at 27 °C day/24 °C night, under a 12 h light and 12 h dark cycle, with 90% relative humidity. Within 0, 12, 24, 48, 72, and 144 h post-inoculation (hpi), petals were harvested. Three biological replicates were conducted for each treatment. The long and short diameters of the lesion spot were ascertained during each time point, and the lesion area was calculated using the formula S = ¼ × 3.14 × a × b (where S represents the lesion area, and a and b are the long and short diameters of the lesion spot, respectively).

### 4.3. RNA Extraction, Library Construction, RNA-Seq, and Differentially Expressed Genes (DEGs) Analysis

Petals were carefully collected at 12 and 144 hpi, and frozen immediately using liquid nitrogen. Total RNA was extracted from each sample using Trizol reagent (Thermo Fisher Scientific, Pleasanton, CA, USA). RNA integrity and the absence of RNA degradation and contamination were assessed using agarose gelelectrophoresis (1%). RNA concentration and purity were determined using NanoDrop 2000 (Thermo Fisher Scientific, Wilmington, NC, USA), while RNA integrity was assessed using the RNA Nano 6000 Assay Kit of the Agilent Bioanalyzer 2100 system (Agilent Technologies, Santa Clara, CA, USA). Sequencing libraries were generated utilizing the NEBNext UltraTM RNA Library Prep Kit for Illumina (New England Biolabs, Inc., Beijing, China) following manufacturer recommendations, and index codes were added to attribute sequences to each sample. The libraries were sequenced by the Beijing Biomarker Biotechnology Company using an Illumina HiSeq 2000 platform, generating approximately 6.14 Gb of clean reads per sample. The RNA-Seq data was generated from three biological replicates. The adaptor sequences and low-quality sequence reads were removed from the data sets. Raw sequences were transformed into clean reads after data processing. These clean reads were mapped to the *Rhododendron simsii* reference genome (http://rhododendron.plantgenie.org/ (accessed on 8 February 2022)) using the Hisat2 tools [60]. Then, the matched reads were assembled into transcripts using StringTie tools for subsequent analysis [61]. Differential expression analysis between infected and mock-inoculated petals was conducted using the DESeq R package (1.10.1). The resulting *p* values were adjusted for multiple testing errors using the Benjamini and Hochberg’s method to control for the false discovery rate (FDR). Genes were identified as differentially expressed when the FDR was less than 0.05 and the absolute fold change (FC) was greater than or equal to 1.5 between treatments.

### 4.4. Gene Ontology (GO) and Kyoto Encyclopedia of Genes and Genomes (KEGG) Pathway Enrichment Analysis

KEGG pathway enrichment and GO enrichment analysis of DEGs was performed using the BMKCloud platform.

### 4.5. Quantitative Real-Time PCR Analysis

To validate the RNA-Seq results, certain annotated genes were selected for quantitative real-time PCR (qRT-PCR). Total RNA was extracted from the samples using the OmniPlant RNA Kit (CW2598S, Cwbiotech, Beijing, China) and reverse transcribed utilizing RevertAid Reverse Transcriptase (EP0441, Thermo Scientific, Beijing, China), following the manufacturer’s protocol. The primer sequences for qRT-PCR are listed in Appendix A, with *EF1α* as the internal control [62]. qRT-PCR was performed utilizing an ABI real-time PCR system (Q1, ABI, Los Angeles, CA, USA). A total of 5 μL 2 × PerfectStart Green qPCR SuperMix (AQ601-04, Transgen Biotech, Beijing, China) was added to the reaction, based on the manufacturer’s instructions. The 2^−ΔΔCt^ method [63] was used to calculate the relative expression levels of the selected genes, normalized to the expression levels of the *EF1α* gene, based on cycle threshold values.

### 4.6. Determination of Hydrogen Peroxide Content and DAB Staining

The hydrogen peroxide (H_2_O_2_) content in the petals was determined using the method described by Velikova et al. (2000) [64]. Petal tissues weighing 300 mg were homogenized in an ice bath with 2 mL of 0.1% TCA (*w*/*v*). The homogenate was then centrifuged at 12,000× *g* for 20 min at 4 °C and 0.5 mL of the supernatant was mixed with 0.5 mL of potassium phosphate buffer (10 mM, pH 7.0) and 1 mL of KI (1 M). The mixture was kept in the dark for 1 h, and the absorbance of the supernatant was measured at 390 nm. The H_2_O_2_ content was calculated using a standard curve. The 3,3-diaminobenzidine (DAB) staining was performed using a staining kit (G1022, Service Biotech, Shanghai, China), following the manufacturer’s instructions.

### 4.7. Metabolites Extraction, UHPLC–MS Analysis, and Data Preprocessing

Petals were collected at 12 and 144 hpi and immediately frozen in liquid nitrogen. The freeze-dried samples were homogenized with a mixer mill for 30 s at 60 Hz. An accurately weighed 50 mg aliquot from each sample was transferred to an Eppendorf tube, and 700 μL of a precooled (at −40 °C) extract solution (methanol/water = 3:1, containing internal standard) was added. After vortexing for 30 s, the samples were homogenized at 40 Hz for 4 min and sonicated for 5 min in an ice-water bath. This homogenization and sonication process was repeated three times. The resulting samples were extracted overnight at 4 °C on a shaker. The samples were then centrifuged at 13,800× *g* for 15 min at 4 °C, and the supernatant was carefully filtered through a 0.22 μm microporous membrane. The resulting supernatants were diluted 20 times with a methanol/water mixture (*v*:*v* = 3:1, containing internal standard). The sample was vortexed for 30 s and transferred to 2 mL glass vials. To ensure quality control, 20 μL from each sample was pooled to create QC samples.

The UHPLC separation was conducted utilizing an EXIONLC System. The mobile phase A was composed of 0.1% formic acid in water, with mobile phase B consisting of acetonitrile. The column temperature was held at 40 °C, while the auto-sampler temperature was maintained at 4 °C, and the injection volume was set at 2 μL. The assay development was performed using a Sciex QTrap 6500+ (Sciex Technologies, Framingham, MA, USA) instrument with typical ion source parameters, including an ion spray voltage of +5500/−4500 V, a curtain gas pressure of 35 psi, an ion source temperature of 400 °C, an ion source gas 1 pressure of 60 psi, an ion source gas 2 pressure of 60 psi, and DP set at ±100 V.

The SCIEX Analyst Workstation software (Version 1.6.3) was utilized for MRM data acquisition and processing. The MS raw data (.wiff) files were converted to the TXT format utilizing the MSc converter. Peak detection and annotation were executed using an in-house R program and database. MetaboAnalyst 5.0 was employed for multivariate analyses, including orthogonal projections to latent structures-discriminant analysis (OPLS-DA), principal component analysis (PCA), and KEGG pathway enrichment. The VIP value of the first principal component in OPLS-DA analysis was acquired, and metabolites with a VIP > 1 and a *p*-value ˂ 0.05 (Student’s *t*-test) were considered to be significantly altered metabolites between the groups.

### 4.8. Correlation Analysis between Trancriptome and Metabolome

The correlation analysis between transcriptome and metabolome was performed by the R.4.2.0 vegan package with Pearson algorithm to examine the potential relationship between differently expressed genes and significantly changed metabolites (*p* < 0.05).

### 4.9. Statistical Analysis

The data are expressed as mean ± standard deviation (SD), and were analyzed using either a one-way analysis of variance (ANOVA) or SNK *t*-test. Analysis was carried out in at least three replicates for each sample, and a *p* < 0.05 was considered statistically significant.

## 5. Conclusions

*Rhododendron* ‘Xiaotaohong’ was utilized as a plant system to study the early and late response mechanisms of petals to fungal infection. Petal blight was observed in MR-9-infected petals. Transcriptomics analysis revealed that MR-9 infection induced a large number of gene transcriptions, which were primarily involved in the plant–pathogen interaction pathway, performing stress-response functions. In this pathway, the accumulation of H_2_O_2_ and the expression of defense-related genes may contribute to the early and late defense responses, while the accumulation of DAMs may play an important role in the late defense response of *Rhododendron* ‘Xiaotaohong’ petals to MR-9 infection. Our next step involves comparing the differentially expressed genes enriched in KEGG and GO, using the blast method, in nucleotide or protein databases. We will focus on a specific gene that responds to pathogen infection in *Rhododendron* ‘Xiaotaohong’ petals and further investigate its gene family. Additionally, we will analyze the biological functions of this gene through biochemical or molecular biology methods.

## Figures and Tables

**Figure 1 ijms-24-12695-f001:**
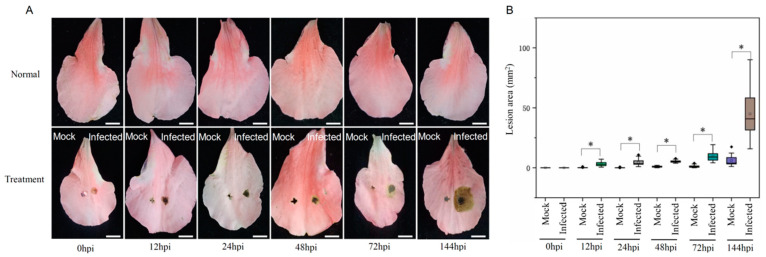
The petal blight disease symptoms on the petal in *Rhododendron* ‘Xiaotaohong’ infected by the blight fungus (MR-9) in a greenhouse. (**A**) The representative petals were obtained in 0, 12, 24, 48, 72, and 144 hpi of strain MR-9, bars = 6 mm. (**B**) The lesion area was measured in 0, 12, 24, 48, 72, and 144 hpi of strain MR-9. * represents the significant difference between infected and mock petals (*p* < 0.05).

**Figure 2 ijms-24-12695-f002:**
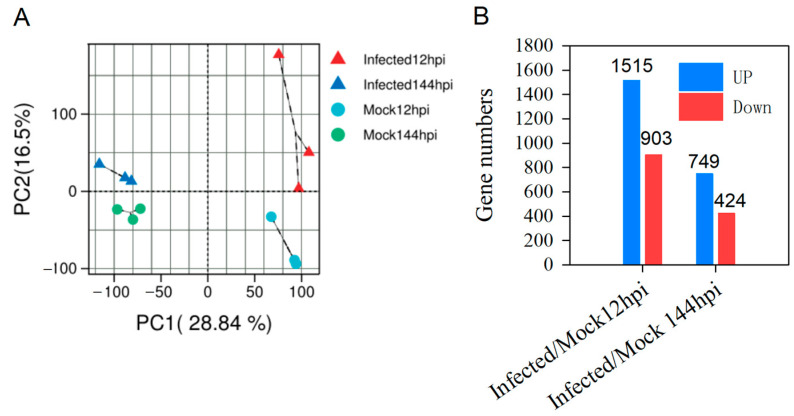
Summary of transcriptome data and differentially expressed genes (DEGs) in the *Rhododendron* ‘Xiaotaohong’ petal response to blight fungus infection. (**A**) Principal component analysis (PCA) of all detectable genes in the petals infected with MR-9 or H_2_O in 12 and 144 h. (**B**) The numbers of up-regulated and down-regulated DEGs infected by MR-9 in 12 and 144 h compared with the mock treatment.

**Figure 3 ijms-24-12695-f003:**
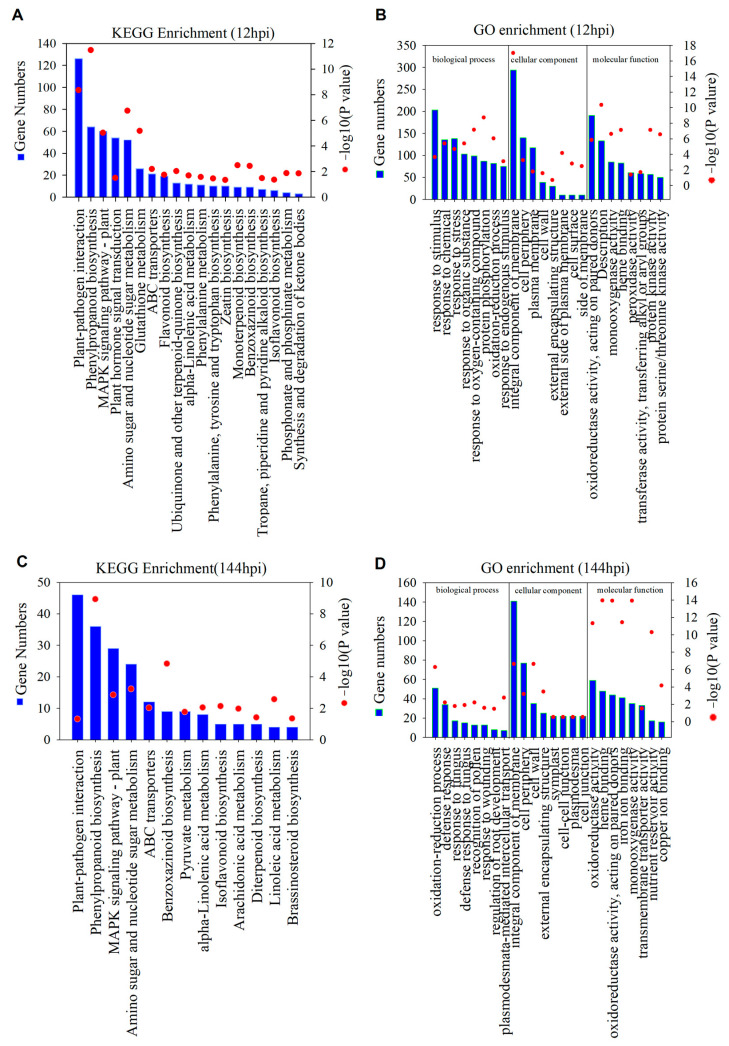
The *Kyoto Encyclopedia of Genes and Genomes* (KEGG) and *Gene Ontology* (GO) enrichment analysis of DEGs: (**A**) enrichment of KEGG annotations of up-regulated DEGs in the petals infected with MR-9 in 12 hpi; (**B**) GO enrichment of up-regulated DEGs in the petals infected with MR-9 in 12 hpi; (**C**) enrichment of KEGG annotations of up-regulated DEGs in the petals infected with MR-9 in 144 hpi; (**D**) GO enrichment of up-regulated DEGs in the petals infected with MR-9 in 144 hpi.

**Figure 4 ijms-24-12695-f004:**
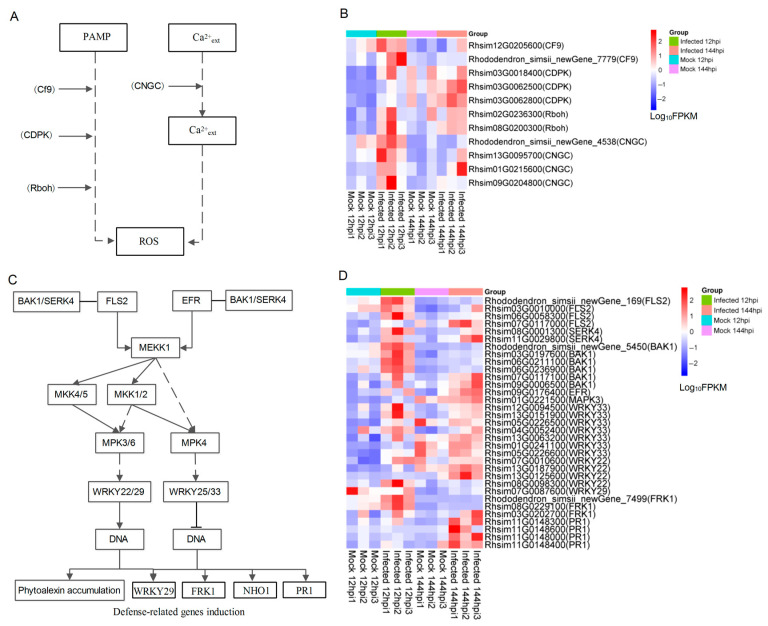
Hydrogen peroxide (H_2_O_2_) and defense-related genes were involved in the response to MR-9 infection in *Rhododendron* ‘Xiaotaohong’ petals. (**A**) ROS production in the plant–pathogen interaction pathway in KEGG enrichment. (**B**) The DEGs involved in the H_2_O_2_ production of the MR-9 infection. (**C**) Defense-related genes induction in the plant–pathogen interaction pathway in KEGG enrichment. (**D**) The DEGs involved in the defense-related genes induction of the MR-9 infection.

**Figure 5 ijms-24-12695-f005:**
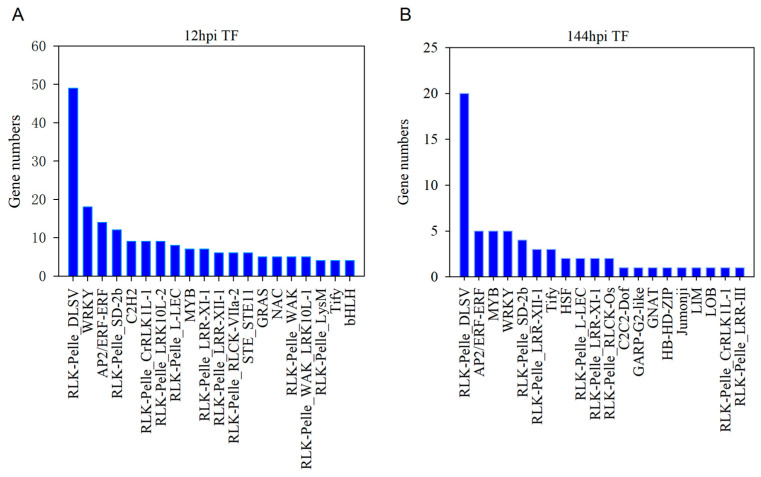
The predicting transcription factors (TFs) encoded by DEGs in 12 hpi petals: (**A**) the TFs encoded by up-regulated DEGs in 12 hpi; (**B**) the TFs encoded by up-regulated DEGs in 144 hpi.

**Figure 6 ijms-24-12695-f006:**
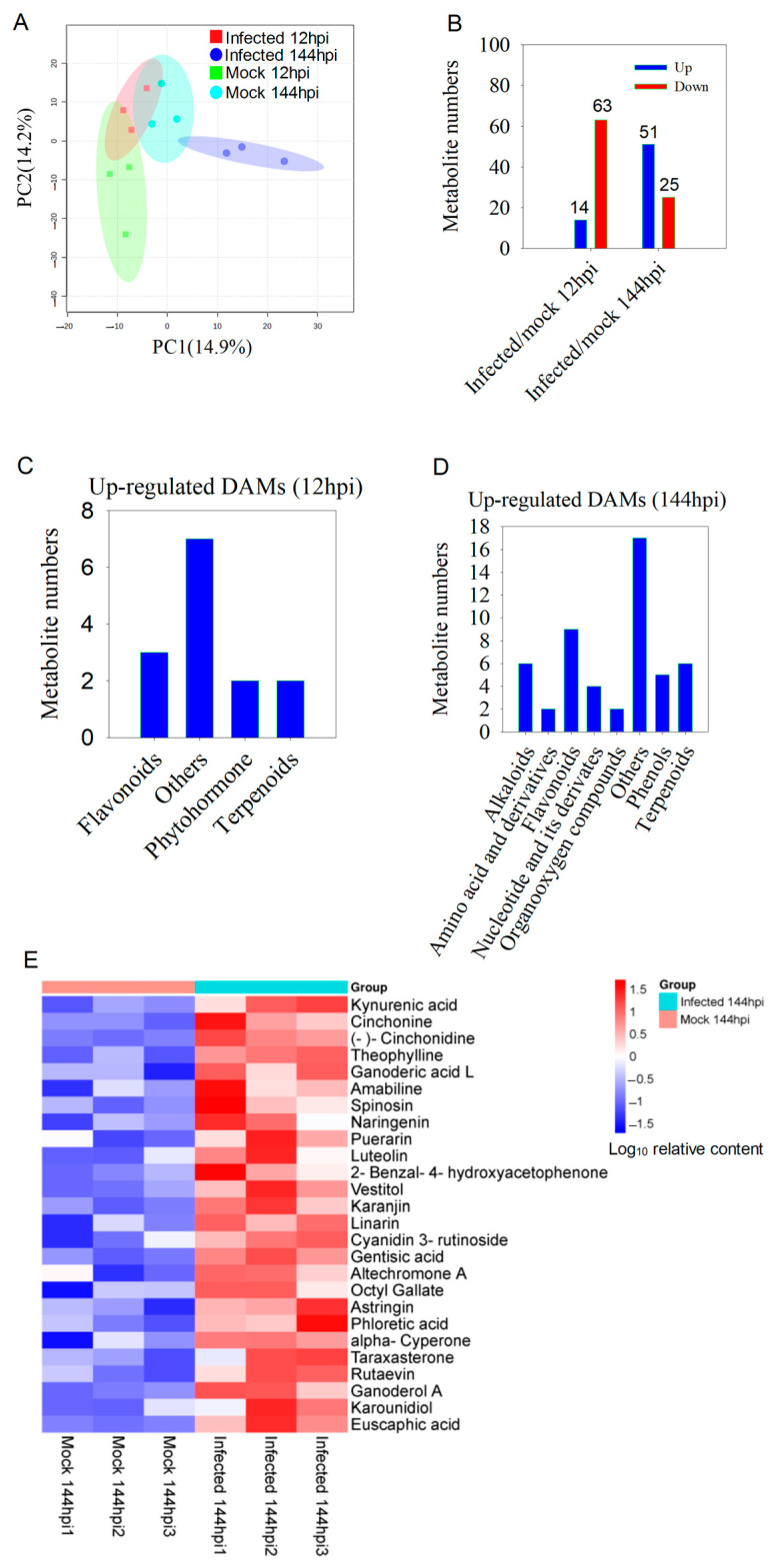
Summary of metabolome data, differentially abundant metabolites (DAMs), and DAM pathway analysis in the *Rhododendron* ‘Xiaotaohong’ petal response to blight fungus: (**A**) principal component analysis of all detectable metabolites in the petals infected with MR-9 or H_2_O in 12 hpi and 144 hpi; (**B**) the numbers of up-regulated and down-regulated DAMs infected by MR-9 in 12 hpi and 144 hpi compared with the mock treatment; (**C**) the pathway enriched by up-regulated DAMs in 12 hpi; (**D**) the pathway enriched by up-regulated DAMs in 144 hpi; (**E**) the accumulation of DAMs infected by MR-9 in 144 hpi.

**Figure 7 ijms-24-12695-f007:**
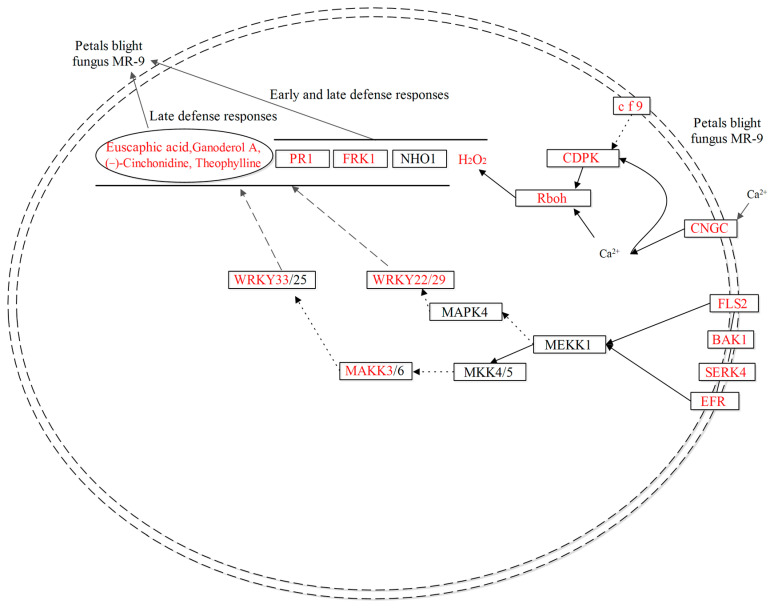
Schematic views of *Rhododendron* ‘Xiaotaohong’ petal response to blight fungus MR-9 from the integrated metabolome and transcriptome analysis. The names in red or black indicate genes or metabolites with up-regulated expression or non-identification, respectively. Boxes and ellipses indicate genes and metabolites, respectively. PR1, FRK1 and H_2_O_2_ might involve the early and late defense responses. Euscaphic acid, ganoderol A, theophylline, and (−)-Cinchonidine might involve the late defense responses.

**Table 1 ijms-24-12695-t001:** The up-regulated alkaloids, flavonoids, phenols, and terpenoids and tentative phytoalexin according to the published literature.

Compounds	Formula	Q1 (Da)	Q3 (Da)	M W (Da)	Class I	IM	RT	Mock 144 hpi	Infected 144 hpi	VIP	*p* Value	Fold Change	Log2FC	References
Kynurenic acid	C10H7NO3	190	144.0	189.043	Alkaloids	+	4.36	3.43 × 10^−5^	6.97 × 10^−5^	1.80	0.00	2.03	1.02	
Cinchonine	C19H22N2O	295.2	79.1	294.173	Alkaloids	+	11.99	1.20 × 10^−4^	4.40 × 10^−4^	1.84	0.01	3.65	1.87	
(−)-Cinchonidine	C19H22N2O	295.2	81.1	294.173	Alkaloids	+	11.99	1.70 × 10^−4^	7.84 × 10^−4^	1.92	0.00	4.60	2.20	
Theophylline	C7H8N4O2	181.1	124.0	180.065	Alkaloids	+	3.92	6.88 × 10^−6^	3.01 × 10^−5^	1.77	0.00	4.38	2.13	
Ganoderic acid	C30H46O8	535.3	499.3	534.319	Alkaloids	+	5.82	5.51 × 10^−6^	1.84 × 10^−5^	1.29	0.01	3.34	1.74	
Amabiline	C15H25NO4	284.2	122.1	283.178	Alkaloids	+	12.99	6.48 × 10^−4^	1.24 × 10^−3^	1.56	0.04	1.92	0.94	
Spinosin	C28H32O15	607.5	606.8	608.174	Flavonoids	+	5.79	7.70 × 10^−5^	1.18 × 10^−4^	1.68	0.04	1.53	0.62	
**Naringenin**	C15H12O5	273.3	152.8	272.069	Flavonoids	+	8.86	5.85 × 10^−5^	1.98 × 10^−4^	1.50	0.03	3.39	1.76	[28]
Puerarin	C21H20O9	417.4	416.8	416.111	Flavonoids	+	4.89	8.46 × 10^−4^	1.13 × 10^−3^	1.59	0.04	1.34	0.42	
**Luteolin**	C15H10O6	287.1	153.0	286.048	Flavonoids	+	8.18	1.35 × 10^−5^	4.74 × 10^−5^	1.59	0.04	3.49	1.80	[29]
2-Benzal-4-hydroxyacetophenone	C15H12O2	225.1	103.1	224.084	Flavonoids	+	5.1	5.31 × 10^−5^	1.30 × 10^−4^	1.74	0.03	2.46	1.30	
**Vestitol**	C16H16O4	273.1	123.0	272.105	Flavonoids	+	6.51	4.00 × 10^−5^	7.97 × 10^−5^	1.85	0.00	1.99	0.99	[30]
**Karanjin**	C18H12O4	293.1	105.0	292.074	Flavonoids	+	11.67	6.02 × 10^−5^	2.34 × 10^−4^	1.83	0.00	3.89	1.96	[31]
Linarin	C28H32O14	593.2	285.1	592.179	Flavonoids	+	7.65	7.34 × 10^−6^	2.53 × 10^−5^	1.47	0.01	3.45	1.78	
Cyanidin 3-rutinoside	C27H30O15	595.2	287.1	594.158	Flavonoids	+	6.08	2.02 × 10^−3^	4.50 × 10^−3^	1.60	0.01	2.22	1.15	
**Gentisic acid**	C7H6O4	153.0	108.0	154.027	Phenols	−	3.62	3.82 × 10^−5^	1.21 × 10^−4^	1.88	0.00	3.17	1.66	[32]
Altechromone A	C11H10O3	191.1	173.1	190.063	Phenols	+	8.25	9.85 × 10^−6^	1.65 × 10^−5^	1.59	0.03	1.68	0.75	
Octyl Gallate	C15H22O5	283.2	153.0	282.147	Phenols	+	12.97	5.06 × 10^−4^	8.31 × 10^−4^	1.54	0.03	1.64	0.72	
Astringin	C20H22O9	407.1	245.1	406.126	Phenols	+	11.6	1.81 × 10^−5^	4.75 × 10^−5^	1.61	0.01	2.61	1.39	
Phloretic acid	C9H10O3	167.1	121.1	166.063	Phenols	+	2.84	1.31 × 10^−3^	2.19 × 10^−3^	1.73	0.02	1.67	0.74	
alpha-Cyperone	C15H22O	219.2	67.1	218.167	Triterpenoids	+	12.2	1.47 × 10^−4^	2.33 × 10^−4^	1.65	0.01	1.58	0.66	
Taraxasterone	C30H48O	425.4	407.4	424.371	Triterpenoids	+	13.85	1.29 × 10^−4^	2.05 × 10^−4^	1.65	0.03	1.58	0.66	
Rutaevin	C26H30O9	487.2	469.2	486.189	Triterpenoids	+	12.58	7.13 × 10^−5^	1.84 × 10^−4^	1.73	0.01	2.58	1.37	
Ganoderol A	C30H46O2	439.4	421.3	438.350	Triterpenoids	+	11.31	1.63 × 10^−5^	1.39 × 10^−4^	1.84	0.00	8.51	3.09	
Karounidiol	C30H48O2	441.4	405.4	440.365	Triterpenoids	+	10.55	2.79 × 10^−5^	6.62 × 10^−5^	1.61	0.04	2.37	1.24	
** Euscaphic acid **	C30H48O5	489.4	425.3	488.350	Triterpenoids	+	12.35	3.05 × 10^−6^	6.68 × 10^−5^	1.75	0.02	21.88	4.45	[33]

Note: + and − indicate positive and negative ion model, respectively. Bold font indicates that the metabolites have been reported as a phytoalexin. Red metabolites might be suggested as putative phytoalexins in Rhododendron petals.

## Data Availability

The data presented in this study are available on request from the corresponding author.

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
