# Peer review of "Early and Late Transcriptomic and Metabolomic Responses of *Rhododendron* ‘Xiaotaohong’ Petals to Infection with *Alternaria* sp."

_ijms, 2023, doi:10.3390/ijms241612695_

Round 1

Reviewer 1 Report

The paper is generally fine with me. Things to be corrested are below.

Lines 73-77 Sound like coumarins is the infection site. Please change the word order.

L 102 unpunctured petals? –please specify

L178 Fig 4A does not show ROS production during the infection, it show some pathways where DEGs take part of. Rewrite.

l 262 negtive

Table 1 Mock 144hpi Infected 144hpi VIP P value columns The values need be expressed with the number of digits matching the accuracy of the measurements. Also, they all need to be in the same format. For instance, if you measure something with 1% accuracy, express it as 1.27 , not 1.2756492876

L 275 either after or post, not both. Area largest at 144 hpi. Does it grow later or not?

L 297 of must be to

Check your ref list! For example, in line 314 you cite two papers, and neither is on corn nor tomato!

L 361 non-expression? Maybe down-regulated expression?

L 372 give full species name

Singular/plural usage check everywhere

Correct word order in places. Put complement and subject together

check for typos

Author Response

Response to reviewer’s comments on 2524285

Reviewer #1:

General comments:

  1. The paper is generally fine with me. Things to be corrested are below.

Authors’ responses: Thank you very much for your careful reading and professional evaluations on our manuscript.

Specific comments:

  1. Reviewer’s comments:

Lines 73-77 Sound like coumarins is the infection site. Please change the word order.

Authors’ responses: Thank you very much for your careful reading and professional advices. We have revised the sentence “Upon pathogen infection, the accumulation of secondary metabolites occurs in the tissues surrounding the infection site, such as coumarins, flavonoids, terpenes, and nitrogen/sulfur compounds [13,14], possessing the ability to eliminate, hinder the growth, and prevent the spread of pathogens [15], thus effectively reducing the impact of diseases caused by pathogens.” in line 73-77 as “Upon pathogen infection, the accumulation of secondary metabolites occurs in the tissues surrounding the infection site (such as flavonoids, terpenes, and nitrogen/sulfur compounds) or in infection site (such as coumarins)[13,14], possessing the ability to eliminate, hinder the growth, and prevent the spread of pathogens [15], thus effectively reducing the impact of diseases caused by pathogens.” Please see our new revised MS.

  1. Reviewer’s comments:

L 102 unpunctured petals? –please specify.

Authors’ responses: Thank you very much for your careful reading. We have revised the sentence “The Rhododendron ‘Xiaotaohong’ petals remained healthy for a period of 144 h without any treatment, as demonstrated in Figure 1A” in line 102 as “The Rhododendron ‘Xiaotaohong’ unpunctured petals remained healthy for a period of 144 h without any treatment, as demonstrated in Figure 1A.” Please see our new revised MS.

  1. Reviewer’s comments:

L178 Fig 4A does not show ROS production during the infection, it show some pathways where DEGs take part of. Rewrite.

Authors’ responses: Thank you very much for your professional advices. We have rewritten the sentence “Figure 4A displays that reactive oxygen species (ROS) production and the expression of defense-related genes were identified as important responses to fungal infection in this pathway.” in line 178 as “Figure 4A displayed the pathway of reactive oxygen species (ROS) production and some identification of DEGs were involved in fungal infection in the Rhododendron ‘Xiaotaohong’ petals”. Please see our new revised MS.

  1. Reviewer’s comments:

L 262 negtive.

Authors’ responses: Thank you very much for your careful reading. We have revised the word “negtive” in line 262 as “negative”. Please see our new revised MS.

  1. Reviewer’s comments:

Table 1 Mock 144hpi Infected 144hpi VIP P value columns The values need be expressed with the number of digits matching the accuracy of the measurements. Also, they all need to be in the same format. For instance, if you measure something with 1% accuracy, express it as 1.27 , not 1.2756492876

Authors’ responses: Thank you very much for your careful reading and professional advices. According to your suggestion, we have revised Table 1. Please see the Table 1 in our new revised MS.

  1. Reviewer’s comments:

L 275 either after or post, not both. Area largest at 144 hpi. Does it grow later or not?

Authors’ responses: Thank you very much for your careful reading and professional advices. Because we only observe the lesions during 144 hours post-infection, so, we have revised the sentence “Petal blight caused by the blight fungus MR-9 was observed in Rhododendron ‘Xiaotaohong’ petals after 12 hours post-infection (hpi), with the lesion area being at its largest at 144 hpi (Figure 1)” in line 275 as “Petal blight caused by the blight fungus MR-9 was observed in Rhododendron ‘Xiaotaohong’ petals 12 hours post-infection (hpi), with the lesion area being at its largest at 144 hpi during the 144 hpi (Figure 1)”. Please see our new revised MS.

  1. Reviewer’s comments:

L 297 of must be to Check your ref list! For example, in line 314 you cite two papers, and neither is on corn nor tomato!

Authors’ responses: Thank you very much for your professional advices and careful reading. We have revised the sentence “In the current study, four up-regulated genes encoding CNGC and three up-regulated genes encoding CDPK were activated after fungal infection (Figure 4), consistent with previous reports in plants [33,34,35], and they were widely involved in the resistance of fungus [35]” in line 297 as “In the current study, four up-regulated genes encoding CNGC and three up-regulated genes encoding CDPK were activated after fungal infection (Figure 4), consistent with previous reports in plants [33,34,35]”. Please see our new revised MS. In addition, we have revised the sentence “Comparable observations have been reported in corn and tomato [36,38]” in line 314 as “Comparable observations have been reported in Medicago truncatula and rice [36,38]”. Please see our new revised MS.

  1. Reviewer’s comments:

L 361 non-expression? Maybe down-regulated expression?

Authors’ responses: Thank you very much for your professional advices and careful reading. The black name here is an unidentified gene or metabolite. We have revised the word “non-expression” as the word “non-identification”. Please see our new revised MS.

  1. Reviewer’s comments:

L 372 give full species name.

Authors’ responses: Thank you very much for your and careful reading. We have revised the full species name in line 372. Please see our new revised MS. The Alternaria alternate sp. strain (MR-9), which is responsible for petal blight disease, was isolated from R. delavayi exhibiting petal blight disease [55] and is presently maintained in the Key Laboratory of Plant Physiology and Development Regulation.

Reviewer 2 Report

I have completed reviewing the manuscript entitled “Early and late transcriptomic and metabolomic responses of Rhododendron ‘Xiaotaohong’ petals to infection with Alternaria sp.”. The study aims to investigate key factors involved in the response of petals of Rhododendron ‘Xiaotaohong’ to Alternaria sp. infection using transcriptomics and metabolomics. The topic of the manuscript is really interesting and the approach used is proper in order to provide valuable insights into the underlying mechanisms of petal response to fungal infection. The study was well designed and all analyses were done properly with the results supporting the intended research objectives. However, some important points in the methodology and results need to be adressed.

The major comment is about the transcriptome annotation: the Rhododendron simsii reference genome was used to map the clean reads resulting from RNA sequencing. This approach resulted in a good KEGG and GO enrichment (different pathways and terms significantly enriched), but the singular gene annotation is not sufficiently detailed in order to distinguish different genes belonging to the same family, maybe because the low resolution annotation of the genome used as reference. Supplementary tables such as Tables S6 and S7, in which DEGs enriched in the different pathways are listed, report general pathways annotation and genes belonging to the same pathway have the same name. Gene families include different genes with different functions and I think it’s fundamental to have a precise gene annotation that identify the single gene. This approach also allow to compare and reinforce the results with those obtained in other species and experiments. I suggest to better identify the candidate genes described in the discussion section by blasting them in nucleotidic or aminoacidic databases, looking for orthologs.

A rich literature reports mechanisms of plant response to fungus infection (e.g. 10.3390/plants10020244, 10.3390/ijms22020882, 10.3390/biology11050761, https://doi.org/10.1080/17429145.2023.2243097), focusing on hormone metabolism and regulation, ROS generation, secondary metabolites production such as pigments important in flower color, transcription factors and PTI and HR mechanisms, with an high detail of the specific genes involved. I suggest to use these information to better support the discussion section and the rich and interesting metabolite analysis performed, regarding all the aspects above mentioned.

In Figure 2 the different samples clustered separately when considering FPKM. The major distance between mock and infected samples at 12hpi compared with the mock and infected samples at 144hpi suggest a strong early response of the plant to the pathogen. How do you explain the difference between mock 12hpi and mock 144hpi? Have you considered a wounding effect during time, also considering that in figure 6A mock 144hpi is similar to infected 12hpi in terms of metabolites composition?

Line 144 and Figure S1: I consider R2=0.55 too low for an RNAseq experiment validation. Can you improve this correlation?

Line 161: the results are referred to figure 3C, not 3B. Please correct

Section 2.3: Why only the up-regulated pathway and terms are described. I think that also the down-regulation contain interesting information. Please give explanations.

Line 178: the results are referred to figures 3A and 3C. Please correct.

Figure 6A: please improve the quality and the resolution of the figure. Check if the legend corresponds with the graph.

Section 2.5: considered the strong transcriptomic response at the early stage of infection, have you considered to correlate metabolites and DEGs also at 12hpi?

Author Response

Response to reviewer’s comments on 2524285

Reviewer #2:

General comments:

  1. I have completed reviewing the manuscript entitled “Early and late transcriptomic and metabolomic responses of Rhododendron‘Xiaotaohong’ petals to infection with Alternariasp.”. The study aims to investigate key factors involved in the response of petals of Rhododendron ‘Xiaotaohong’ to Alternaria sp. infection using transcriptomics and metabolomics. The topic of the manuscript is really interesting and the approach used is proper in order to provide valuable insights into the underlying mechanisms of petal response to fungal infection. The study was well designed and all analyses were done properly with the results supporting the intended research objectives. However, some important points in the methodology and results need to be addressed.

Authors’ responses: Thank you very much for your professional evaluations and careful reading on our manuscript.

Specific comments:

  1. Reviewer’s comments:

The major comment is about the transcriptome annotation: the Rhododendron simsii reference genome was used to map the clean reads resulting from RNA sequencing. This approach resulted in a good KEGG and GO enrichment (different pathways and terms significantly enriched), but the singular gene annotation is not sufficiently detailed in order to distinguish different genes belonging to the same family, maybe because the low resolution annotation of the genome used as reference. Supplementary tables such as Tables S6 and S7, in which DEGs enriched in the different pathways are listed, report general pathways annotation and genes belonging to the same pathway have the same name. Gene families include different genes with different functions and I think it’s fundamental to have a precise gene annotation that identify the single gene. This approach also allow to compare and reinforce the results with those obtained in other species and experiments. I suggest to better identify the candidate genes described in the discussion section by blasting them in nucleotidic or aminoacidic databases, looking for orthologs.

Authors’ responses: Thank you very much for your professional advice. Your suggestion is very good. Our next step is to compare the differentially expression genes enriched in KEGG and GO using the blast method in nucleotide or protein databases. A single gene that responds to pathogen infection in Rhododendron ‘Xiaotaohong’ petals will be further explored from gene families. Then, their biological functions will be analyzed through biochemical or molecular biology methods. We have already explained in the conclusion section. Please see conclusion section in our new revised MS.

  1. Reviewer’s comments:

A rich literature reports mechanisms of plant response to fungus infection (e.g. 10.3390/plants10020244, 10.3390/ijms22020882, 10.3390/biology11050761, https://doi.org/10.1080/17429145.2023.2243097), focusing on hormone metabolism and regulation, ROS generation, secondary metabolites production such as pigments important in flower color, transcription factors and PTI and HR mechanisms, with an high detail of the specific genes involved. I suggest to use these information to better support the discussion section and the rich and interesting metabolite analysis performed, regarding all the aspects above mentioned.

Authors’ responses: Thank you very much for your professional advices and providing more references. We have cited three of the above mentioned literature into the discussion section (10.3390/plants10020244, 10.3390/ijms22020882, and 10.1080/17429145.2023.2243097). We believe that with the support of these literature, the discussion section will be more persuasive. Please see references section in our new revised MS.

  1. Reviewer’s comments:

In Figure 2 the different samples clustered separately when considering FPKM. The major distance between mock and infected samples at 12hpi compared with the mock and infected samples at 144hpi suggest a strong early response of the plant to the pathogen. How do you explain the difference between mock 12hpi and mock 144hpi? Have you considered a wounding effect during time, also considering that in figure 6A mock 144hpi is similar to infected 12hpi in terms of metabolites composition?

Authors’ responses: Thank you very much for your careful reading and professional advices. As mentioned in the method section, petals were punctured with a sterilized 2 mm carving knife. It is precisely because we believe that puncture may produce a stress on petals, which may induce gene expression and metabolite accumulation. We set up two control treatments, which is mock 12hpi and mock 144hpi. We believe that the difference between mock 12hpi and mock 144hpi in figure 2 is caused by puncture. We have added the sentence “In addition, the difference between mock 12hpi and mock 144hpi was observed (Figure 2A), suggesting that it is caused by puncture.” to discussion section. Please see our new revised MS. The shorter distance between mock 144hpi and infected 12hpi in terms of metabolites composition based on principal component analysis were found in figure 6A, we believe that the changes in metabolites caused by fungal infection in the short term may be similar to those caused by long-term puncture stress.

  1. Reviewer’s comments:

Line 144 and Figure S1: I consider R2=0.55 too low for an RNAseq experiment validation. Can you improve this correlation?

Authors’ responses: Thank you very much for your careful reading and professional advices. The samples used for PCR analysis and RNA sequencing are from the same batch. After PCR analysis, these samples have been fully used. Therefore, we do not have any samples for PCR analysis in the short term.

  1. Reviewer’s comments:

Line 161: the results are referred to figure 3C, not 3B. Please correct.

Authors’ responses: Thank you very much for your careful reading. We have revised the Figure 3B in line 161 as Figure 3C. Please see our new revised MS.

  1. Reviewer’s comments:

Section 2.3: Why only the up-regulated pathway and terms are described. I think that also the down-regulation contain interesting information. Please give explanations.

Authors’ responses: Thank you very much for your careful reading and professional advices. We also analyzed the KEGG and GO enrichment of down-regulated DEGs in the petals by infected pathogen, and found that after pathogen infection of Rhododendron ‘Xiaotaohong’ petals, the down-regulated DEGs were significantly enriched in the photosynthesis, photosynthesis – antenna proteins, start and sucross metabolism pathways. Therefore, we speculated that the down-regulated DEGs in petals by infected pathogen mainly limit photosynthesis and sucrose metabolism, instead of being involved in the response to pathogens. The majority of up-regulated DEGs were found to be enriched in plant-pathogen interaction pathways according to KEGG enrichment. Therefore, we mainly analyze the enrichment of up-regulated DEGs in KEGG and GO to reveal the responses of Rhododendron ‘Xiaotaohong’ petals to infection with Alternaria sp.

  1. Reviewer’s comments:

Line 178: the results are referred to figures 3A and 3C. Please correct.

Authors’ responses: Thank you very much for your careful reading and professional advices. We have revised the Figure 3A in line 178 as Figure 3A and 3C. Please see our new revised MS.

  1. Reviewer’s comments:

Figure 6A: please improve the quality and the resolution of the figure. Check if the legend corresponds with the graph.

Authors’ responses: Thank you very much for your professional advices. We have redrawn the Figure 6 and improved the quality and the resolution in Figure 6A. In addition, we also carefully checked the legend in Figure 6A. Please see the revised Figure 6 in our new revised MS.

  1. Reviewer’s comments:

Section 2.5: considered the strong transcriptomic response at the early stage of infection, have you considered to correlate metabolites and DEGs also at 12hpi?

Authors’ responses: Thank you very much for your professional advices. We have conducted a correlation analysis between DEGs and metabolites at 12hpi and 144hpi in the third paragraph in section 2.5. The results showed that many up-regulated genes were positively correlated with the accumulation of H2O2 (as a metabolite) at 12 h, gentisic acid, karanjin, ganoderol A, and euscaphic acid at 144 h (Figure S4). Please see section 2.5 in our new revised MS.